# Which Is More Important for Health: Sleep Quantity or Sleep Quality?

**DOI:** 10.3390/children8070542

**Published:** 2021-06-24

**Authors:** Jun Kohyama

**Affiliations:** Tokyo Bay Urayasu Ichikawa Medical Center, Urayasu 279-0001, Japan; j-kohyama@jadecom.or.jp; Tel.: +81-47-351-3101

**Keywords:** restfulness, sleepiness, sleep duration, insufficient sleep syndrome, Pittsburgh Sleep Quality Index (PAQI), Epworth Sleepiness Scale (ESS)

## Abstract

Sleep is one of the basic physiological processes for human survival. Both sleep quantity and sleep quality are fundamental components of sleep. This review looks at both sleep quantity and sleep quality, considering how to manage the complex but probably unavoidable physiological phenomenon of sleep. The need for sleep has marked variations between individuals, in addition to the effects of variable conditions. Studies on sleep quality started later than those on sleep quantity, beginning in 1989 when Ford and Kamerow revealed that insomnia increases the risk of psychiatric disorders. According to the nationwide research team on the quality of sleep (19FA0901), sleep quality is superior to sleep quantity as an index for assessing sleep, and that restfulness obtained through sleep is a useful index for assessing sleep quality. We should pay more attention to obtaining sleep of good quality (restfulness, no sleepiness, no need for more sleep, sufficient objective sleep depth, etc.), although there have not been enough studies on the associations between sleep quality and health or disorders in children and adolescents. Further studies using the deviation from an individual’s optimal sleep quantity may show us another aspect of the effects of sleep quantity on various life issues.

## 1. Introduction

There is no doubt that sleep is one of the basic physiological processes for human survival [1]. However, with increasing economic and social demands, chronic sleep loss has been appreciated [2]. Under such circumstances, for most people in Japan, striving to achieve their best seems to become an important principle, even if this comes at the expense of sleep [3]. At least people in the modern society of Japan seem to be trying to reduce the duration of their sleep on the basis of the superficial idea that they can live effective lives without sleep, although much information on the importance of sleep has accumulated [1]. They want to spend the minimum effective time asleep, because recently people have thought of sleeping time as useless time. In fact, in Japan, sleep duration has decreased by 59 min during the last 50 years [4]. In the USA, the mean sleep duration decreased from 7.40 h in 1985 to 7.18 h in 2004. The percentage of adults in the USA sleeping for 6 h or less increased between 1985 (22.3%) and 2004 (28.6%) [5].

During 1905–2008, the sleep duration of children around the world aged 5–18 years decreased by approximately 0.75 min per night per year [4]. These declines were obvious in Asia (the mean change of minutes per year; −0.50), Canada (−0.73), parts of Europe (−0.92), and the USA (−0.53), while sleep duration increased in Australia (+1.27), Scandinavia (+0.00) and the UK (+0.57) [6]. The sleep durations of preschoolers in Japan, especially nocturnal ones, reduced markedly over a period of nearly 70 years (between 1935–1936 and 2003) [7]. In both studies conducted in 1935–1936 and 2003, bedtime, waking time, and nap duration were asked through a direct interview to each mother. In 1935–1936, the average sleep duration of infants aged 6–11 months was 13.0 h, with a nocturnal sleep duration of 11.3 h, while in 2003 the average was 11.7 h with a nocturnal sleep duration of 10.1 h. For 3-year-old children, the average in 1935–1936 was 11.3 h with a nocturnal sleep duration of 11.0 h, and in 2003 the figures were 11.1 h and 9.7 h. The average sleep duration of 6-year-old children in 1935–1936 was 10.8 h, with the same nocturnal sleep duration, while in 2003 the figure was 10.2 h with a nocturnal sleep duration of 9.8 h. According to the Japan Society of School Health [8], the sleep duration of school pupils in Japan has recently been decreasing. From 1981 to 2016, sleep durations of grades 5 and 6 elementary school (ES) pupils decreased by 17 min in males and 24 min in females. Similarly, those of junior high school (JHS) pupils decreased 40 (male) and 38 (female) minutes. In comparison with 1992, male and female senior high school (SHS) pupils slept 12 (male) and 6 (female) minutes less in 2016. It would not be a mistake to say that the sleep duration of children has recently decreased. Sharma and Kavuru [9] described modern society as a sleep-deprived society, stating that the average sleep duration in modern times is 6.8 h, as opposed to 9 h a century ago, although they gave no citation for this. In modern society, sleep is often not made a priority due to competing interests such as sports, media usage, and so on. Some people believe they can reduce their sleep duration if they can have high quality sleep.

Regarding recent trends in US adult sleep duration, the mean sleep duration showed little change from 2004 to 2012, and the percentage of adults sleeping for 6 h or less (29.2%) also showed little change from 2004 to 2012 [5]. A systematic review [10] on adult sleep duration in 15 countries revealed that sleep duration from the 1960s until the 2000s increased in seven of the countries (Britain, Bulgaria, Canada, France, Korea, the Netherlands, and Poland (range: 0.1–1.7 min per night per year)), decreased in six of the countries (Austria, Belgium, Finland, Germany, Japan, and Russia (range: 0.1–0.6 min per night per year)), and showed inconsistent results for Sweden and the USA. In spite of the aforementioned description of a sleep-deprived society [9], we are unable to confirm decreases in adult sleep duration since there is no data available from the early 1900s. However, it should also be noted that Matricciani et al. described that not adults’ sleep duration but adults’ sleep quality is declining recently [11].

The importance of sleep quality has gained recognition as an important sleep characteristic much later than studies conducted solely on sleep quantity. In 1964, Hammond reported that those who had a sleep duration of 7 h showed the lowest mortality during a 2-year follow up, with increasing death rates on both the shorter and the longer sides of this nadir [12]. Hammond also reported an association between insomnia and mortality in men [12]. Although no detail on the definition of insomnia was given in the report, non-restorative sleep or “not feeling refreshed after sleeping” was described as a symptom of insomnia in the 4th [13] and the 5th [14] editions of the Diagnostic and Statistical Manual of Mental Disorders, respectively. Thus, it may not be wrong to say that Hammond’s insomnia could express a kind of sleep quality. After this study, however, many studies were conducted with a focus on the relationship between sleep duration and both mental and physical disorders, and Ford and Kamerow revealed in 1989 that insomnia increased the risk of psychiatric disorders [15]. At present, there is no definitive definition that can be used to assess sleep quality. In fact, a scale that is commonly used worldwide for assessing sleep quality (the Pittsburgh Sleep Quality Index (PSQI)) is known, but the questions in the PSQI are about “usual” sleep habits during the last month. According to Pilz et al. [16], the PSQI reveals sleep quality on workdays. The PSQI is presumed not to be a perfect scale for measuring sleep quality. Several recent studies [17,18,19,20,21,22,23,24,25,26,27,28,29] have used a simple question to assess sleep quality, and the details of this are introduced in a subsequent section.

This review looks at both sleep quantity and sleep quality, considering how to manage the complex but probably unavoidable physiological phenomenon of sleep. Not enough data on these issues were obtained from children, however, this review tried to accumulate child data on both sleep quantity and sleep quality.

## 2. Sleep Quantity

Sleep quantity decreases from about 18 to 16 h per day in the newborn infant to 7 to 6 h in older individuals. These age-related changes are well-known [30,31]. Although no one knows why these alterations occur and what factors determine them, several figures for the optimal sleep quantity at different ages have been recommended [32,33,34]. Age is not the only determinant of sleep quantity. Sleep quantity is also markedly different among countries. In 17 predominantly Asian and predominantly Caucasian countries/regions [35], sleep quantity was investigated through an Internet survey among children aged from birth to 36 months. This study revealed that total sleep quantity ranged from 11.6 h (Japan) to 13.3 h (New Zealand). The differences seen in sleep quantity may be contributed to cultural differences.

Screen time [36] and extracurricular after-school activities [37] have been identified as factors that contribute to the decrease in sleep quantity. Delayed bedtime was found to decrease the sleep quantity of 3-year-old children [38], and a long waiting time for a late meal may lead to decreased night-time sleep duration in preschoolers [39]. Quante et al. [37] divided the factors that reduce sleep quantity into intrinsic factors and extrinsic ones. In addition to the aforementioned factors, the intrinsic ones include the reduction of the accumulation of sleep pressure building up during the day; early school schedules were one of the latter class of factors. Family lifestyle [40] must also be included among the extrinsic factors. Fukuda et al. [40] reported that children’s bedtime is also determined by delayed waking and meal times. According to a survey conducted during 2016–2018 in 28 public schools (15 ESs, 8 JHSs, and 5 SHSs) [41], the factors significantly associated with a reduction in sleep quantity were longer after-school activity and more sleepiness (in all types of schools), higher grades and longer weekday screen times (in both ES and JHS), irregular dinners, skipping breakfast, longer weekend screen time and better self-reported academic performance (in ES), and higher standardized body mass index (BMI) (in SHS). These associations of short sleep duration with grade-related decline [30,31], screen time [36], after-school activity [37], breakfast skipping [42] and BMI [43] were consistent with previous studies. There are so many issues in modern society that are associated with a decrease in sleep quantity.

## 3. Insufficient Sleep Syndrome

The problems produced by a loss of sleep quantity were well summarized by a 2011 editorial in Sleep Medicine entitled “Give children and adolescents the gift of a good night’s sleep: A call to action” [44], which listed decreased cognitive functioning (e.g., inattention, decreased concentration), poor academic performance, decreased emotional regulation, increased behavior problems, psychopathology, the risk of accidental and automobile crash injuries, long-term deleterious effects on the cardiovascular, immune, various metabolic systems, and obesity.

The International Classification of Sleep Disorders [45] describes the loss of sleep quantity in the second criterion (B) of the diagnostic criteria for insufficient sleep syndrome as “the patient’s sleep time, established by personal or collateral history, sleep logs, or actigraphy, is usually shorter than expected for age”. Loss of sleep quantity is the main issue used in the diagnosis of insufficient sleep syndrome. Among the problems associated with insufficient sleep syndrome, the same standard [45] describes irritability, concentration and attention deficits, reduced vigilance, distractibility, reduced motivation, anergia, dysphoria, fatigue, restlessness, lack of coordination, malaise, increased daytime sleepiness, concentration problems, lowered energy levels, depression, other psychological difficulties, poor work performance, withdrawal from family and social activities, abuse of stimulants, and traffic accidents or injury at work, in addition to behavioral abnormalities in prepubertal children. According to these descriptions, it is assumed to be easy to diagnose an individual as having insufficient sleep syndrome. However, the standard textbook [45] also indicates that there is a poor correlation between subjectively reported sleepiness and multiple sleep latency test-measured sleepiness after sleep deprivation. The textbook mentions that individuals’ different susceptibility to sleep deprivation may produce this poor accordance.

The amount of sleep needed varies from person to person and from night to night [46]. There are a number of factors that influence sleep quantity including: genetic, behavioral, medical, and environmental factors [33]. It is widely recognized that loss of sleep quantity is associated with obesity, elevated blood pressure, and an elevated risk of cardiovascular diseases [44]; however, these results were obtained by comparisons among several sleep duration groups. It is not easy to determine the quantity of sleep that is not associated with the aforementioned health problems in each individual. Indeed, in several recommendations for optimal sleep quantity [32,33,34], relatively wide ranges have been proposed. In a society that ignores sleep, there may be a person who pays attention to the lowest figure in these recommendations because he/she wants to minimize their sleep duration. Most people at least in Japan pay attention to the lowest figure in these recommendations because they want to minimize their sleep duration [3]. However, it should be claimed that there are people who need the highest figure for sleep quantity in these recommendations. It is not only most people but also many clinicians who have not noticed the problems derived from insufficient sleep, because most medical school programs do not have systematic courses on sleep, and fail to show the importance of sleep for various aspects of health [47]. Common complaints by patients about daytime weakness, tiredness, concentration problems, lowered energy levels, poor work performance, traffic accidents, injuries at work, and intellectual troubles may often be misunderstood as the consequences of life stresses such as family or social problems rather than of the more basic cause of sleep insufficiency. It is obvious that medical professionals need to notice that some patients are suffering from sleep disruption including (social) jetlag or shift work. Of course, people should also be educated on the adverse effects of inadequate sleep for their health. Even in a small outpatient clinic for sleep, a considerable number of child patients with insufficient sleep syndrome have been found [48]. In the total of 181 child patients who visited a sleep clinic where the author was associated between August 2012 and March 2017, the most frequent final diagnosis was insufficient sleep syndrome (*n* = 56). The most common complaint in the patients with this syndrome was difficulty in waking in the morning (*n* = 35), followed by feeling sleepy all day (*n* = 12), falling asleep during a school class (*n* = 8), and feeling sick in the morning (*n* = 1). More effort is needed to recognize that the number of children who are suffering from insufficient sleep is not small.

## 4. Sleep Need Index (SNI)

According to the International Classification of Sleep Disorders, Version 3 [45], the diagnostic criteria for insufficient sleep syndrome include sleep duration, and increased daytime sleepiness is said to appear as a result of this syndrome. The hypothesis is that the need for sleep can be expressed as a function of both sleep duration and sleepiness. Thus, the SNI was designed to use the following formula: sleepiness/sleep duration. The larger the value for this index, the larger the sleep need [49]. The SNI calculated in the previously cited survey revealed that SNI values ranged from 0.09 to 1.08, with a mean value of 0.26 and a standard deviation of 0.14. High sleep-need pupils were defined as those whose SNI was 0.26 or higher, while low sleep-need pupils were those whose SNI was less than 0.26. Higher grade, female gender, poor academic performance score, skipping breakfast, longer after-school activity, and higher physical activity were independently associated with high sleep-need pupils. To support adolescents with sleep need, the need for pupils to be instructed to take breakfast and to avoid excessive after-school and physical activities should be emphasized.

## 5. Future Assessment on Sleep Quantity

Since sleep quantity can be measured objectively, albeit in various ways, it can be measured relatively easily. However, there are marked inter-individual differences in the need for sleep. Indeed, young adults with short sleep duration have been reported to be composed of several groups; those who are suffering from insomnia, those who complain of sleep shortage, and those who have no complaints [50]. Sleep quantity might be assessed not by actual duration, but by a deviation from the individual’s own optimal sleep quantity. Therefore, before assessing the association between sleep quantity and other variables, the optimal sleep duration and deviation from an individual’s own optimal sleep quantity should be measured in each person. Further studies using these variables may show us another aspect of the effects of sleep quantity on various life issues. Social jetlag [51] might be a potential indicator to compensate for the inter-individual differences of sleep need.

## 6. Sleepiness

The standard textbook [45] often uses the term “sleepiness” when referring to insufficient sleep syndrome. Why does this subjective word appear so frequently? The assumption could be that objective information on sleep quantity is inadequate to define insufficient sleep syndrome, because the amount of sleep needed has marked inter-individual variations in addition to the effects of variable conditions [45]. According to Komada et al. [51], a low sleep quantity as well as high levels of social jetlag were associated with daytime sleepiness assessed by the Japanese version of the Pediatric Daytime Sleepiness Scale.

Interestingly, marked attention has recently been paid to sleepiness in terms of academic performance or self-regulation. Dewald et al. [52] examined the associations between sleep quantity, sleep quality, sleepiness, and academic performance in three separate meta-analyses, including influential factors (e.g., gender, age, parameter assessment) as moderators. Although all three sleep variables were significantly but modestly related to school performance, sleepiness showed the strongest relationship to academic performance (*r* = −0.133), followed by sleep quality (*r* = 0.096) and sleep quantity (*r* = 0.069). Cohen-Zion and Shiloh [53] investigated the effects of sleep quantity, chronotype, and sleepiness on day-to-day executive abilities and academic performance in a healthy adolescent sample. They found that the strongest predictors of impaired daily executive capacities were evening chronotype and degree of sleepiness, and they concluded that sleep quantity was a weak predictor of executive skills. In the survey among JHS and SHS pupils [54], pupils who recognized themselves as having poor academic performance showed more sleepiness than pupils who recognized themselves as having good (but not very good) academic performance. In all types of schools examined (ES, JHS, and SHS), sleep duration showed no significant differences among the self-reported academic performance categories. Short night-time sleep quantity is known to be a weaker predictor of poor self-regulation than daytime sleepiness among adolescents [55]. Self-regulation in adolescents contributes to a range of positive health and functioning outcomes that have potential long-term implications. Moreover, in school children aged 10–15, daytime sleepiness has been found to be associated with the frequency of neck, shoulder, and back pain [56]. Furthermore, Raine and Venables found that 15-year-old sleepy boys were more likely to be antisocial than non-sleepy boys [57]. From these results, they hypothesized that adolescent sleepiness predisposes an individual to later antisociality. Although they used a very brief and simple assessment of subjective daytime sleepiness, they succeeded in confirming that the well-known link between social adversity and adult crime is partly explained by sleepiness. Their results may have prognostic clinical/social value, and suggest that interventions to reduce sleepiness could be a useful way to prevent future crime.

There are several standardized scales for assessing sleepiness. The Stanford Sleepiness Scale is based on the assumption that fatigue produces sleepiness [58], and the Epworth Sleepiness Scale evaluates the ease with which one falls asleep [59]. However, it should be noted that the test–retest reliability of the Epworth Sleepiness Scale has recently been reported to be poor [60]. Through a single simple question, factors associated with sleepiness were assessed in a survey introduced above [41,54]. Sleepiness was categorized by the frequency with which pupils felt sleepy during class (never, sometimes, often, always). Sleepy pupils were defined as those who selected the choice of either “often” or “always”. By means of a stepwise procedure of multivariable logistic regression analysis, grade, bedtime before school days, non-school day screen time, academic performance, skipping breakfast, waking time on school days, after-school activity, and number of days a week performing habitual exercise except for school lessons (physical activity score), were found to be independently associated with sleepy pupils [61].

Taking these reports into consideration, it would be reasonable to assume that sleepiness is a better potential candidate than sleep quantity for assessing the executive functioning of adolescents, even though it is subjective. We could pay more attention to sleepiness rather than sleep quantity when assessing the executive functioning of pupils. It should also be noted that sleepiness has been regarded as one of the issues that reflect sleep quality [62].

## 7. Sleep Quality

Sleep quality is difficult to define objectively. Even if the polysomnographic recording for a person shows a typical sleep progress chart of a night with a higher rate of deep sleep in the first third of the night, increasing REM sleep and N2 sleep stage duration in the last third of a night, and a low incidence of intermittent waking, the quality of the sleep of the night is defined as poor if the individual was unsatisfied with the night’s sleep [63]. For this reason, we have to define the quality of sleep subjectively.

The Ministry of Health, Labour and Welfare in Japan organized a nationwide research team on the quality of sleep (19FA0901). The author was one of the review board members of the team. The team discussed sleepiness during the day, restlessness, and restfulness as candidates for assessing sleep quality. In 2021, the team reached five conclusions: 1. Sleep quality is a superior sleep index to sleep quantity for assessing sleep; 2. Restfulness obtained through sleep is a useful index for assessing sleep quality; 3. Although PSQI and restfulness are correlated, these two indices are not identical; 4. PSQI includes sleep quantity and insomnia, while restfulness is complementary to either sleep duration, time in bed, or both; and 5. To obtain adequate sleep, people aged 64 or less need a longer sleep duration, whereas those aged 65 or more need a shorter time in bed.

By means of a choice-based conjoint analysis, Ramlee et al. concluded that the top three parameters that determine sleep quality among 17 parameters were total sleep quantity, refreshed feeling upon waking, and the daytime mind state after sleep [64].

Although the study was carried out among older men, Faerman et al. [17] assessed the correlations between subjective sleep quality, determined by 5-point scales for sleep depth and restfulness in the morning, and actigraphic data, as well as heart rate, heart rate variability, and demographic and psychological variables. They found no correlation between the subjective and the objective measures. This result is consistent with a previous study by Kaplan and colleagues [18]. In 3173 men and women aged between 39 and 90, Kaplan et al. assessed the relationship between a morning rating of the prior night’s sleep (sleep depth and restfulness) and polysomnographic and quantitative electroencephalographic descriptors of that single night of sleep. They concluded that objective data contribute little to explaining subjective sleep quality. They also assessed a single night of at-home polysomnographic recording of sleep followed by a set of morning questions concerning the prior night’s sleep quality in older men (*n* = 1024) and women (*n* = 459) [19]. They found that the commonly obtained polysomnographic measures contributed little to the subjective ratings of the prior night’s sleep quality.

Clark et al. [20] reported that the onset of sleep disturbance predicts a subsequent risk of hypertension and dyslipidemia. The onset of impaired sleep in this study included subjective issues such as difficulty in falling asleep, difficulty in maintaining sleep, early morning awakening, and non-restorative sleep. Bin [65] wrote a commentary on this article and summarized the main findings of this report into two statements: 1. Sleep disturbance predicts the occurrence of hypertension and dyslipidemia, and 2. Sleep quality appears a more important risk factor than sleep duration for these disorders.

Falbe et al. assessed restfulness by asking the question “On how many days in the past week have you felt like you needed more sleep?” of 2048 children in grades 4 and 7 [21]. They defined restfulness as having the feeling that more sleep was needed on three days or fewer, based on the notion that perceived restfulness may reflect good sleep quality. Sleeping near a small screen, but not a TV, was associated with a significantly higher prevalence of perceived insufficient restfulness, after adjustment for sleep quantity. This did not vary significantly by grade, gender, physical activity, or race. As regards the reason why TV presence was not related to less restfulness, they speculated that TV sets do not interrupt sleep when turned off. They also demonstrated the association between a longer screen time and less restfulness.

Using “feeling fresh” after waking-up as a representative issue to assess sleep quality of children, the following factors were shown to affect child sleep quality: gene, parent/caregiver, sleep disorders/medical problem, sleep habits/environment/medications, and screen exposure [66].

So far, as mentioned above, sleep depth, restfulness, non-restorative sleep on the prior night, sleepiness, and having the feeling that one needed more sleep on three or fewer days in the past week have been used as subjective measures of sleep quality. Among these, information on restfulness from sleep is routinely obtained using the standard questionnaire in Specific Health Checkups, which is the annual health screening and promotion service organized by the Japanese Ministry of Health, Labour and Welfare [22]. Kaneko et al. [23] investigated the association between restfulness from sleep and subsequent risk of cardiovascular disease using the medical records (obtained from the Japan Medical Data Center) of 1,980,476 individuals (mean age of 45 ± 11 years, male vs. female ratio of 1.49, and mean follow-up period of 1122 ± 827 days) without either prior cardiovascular disease or prior sleep disorders. Among the general population without a prior history of relevant cardiovascular disease, it was found that the incidence of myocardial infarction, angina pectoris, stroke, heart failure, and atrial fibrillation was significantly lower in those with good restfulness from sleep. However, this study did not assess the association of cardiovascular disease with sleep quantity.

## 8. Sleep Quantity and Quality in Patients with Autism Spectrum Disorder and Attention Deficit/Hyperactivity Disorder

Lugo et al. [67] conducted a meta-analysis including 42 studies, and found that both adult autism spectrum disorder (ASD) and attention deficit/hyperactivity disorder (ADHD) patients showed higher sleep onset latency, poorer sleep efficiency, greater number of awakenings during sleep, and a general lower self-perceived sleep quality compared with healthy controls. Another meta-analysis in children and adults with ADHD, reported that short sleep duration was significantly associated with ADHD symptomatology, especially hyperactivity [68]. In ASD and ADHD children, more sleep problems and shorter sleep duration than controls were reported [69]. The same study also found that sleep problems are related to inadequate sleep hygiene in ASD and to evening chronotype in ADHD. Sleep studies on ASD and ADHD should also be encouraged.

## 9. Quantity Versus Quality

Seow et al. [70] studied the association of both sleep quantity and sleep quality with physical and mental disorders in Singaporean adults. Although this study used PSQI as an index to assess sleep quality, it showed that both short sleep duration and poor sleep quality were associated with chronic pain, obsessive compulsive disorders, and mental disorders. In addition, poor sleep quality was found to be associated with major depressive disorder, bipolar disorder, generalized anxiety disorder, and physical disorder. After these analyses, the authors concluded that sleep quality is a more important indicator for psychological and overall health than sleep quantity.

Lao et al. [24] examined associations between sleep measures and the development of coronary heart disease among 2740 adults aged 40 years or above who participated in a Taiwanese cohort. These authors assessed both sleep quantity and sleep quality. Sleep quality was assessed using a 5-point scale. The question was “How do you evaluate your sleep situation last month?”, and the five possible selections were as follows: use of sleeping pills or drugs, difficult to fall asleep, dreamy sleep, can fall asleep but easily awaken, and sleep well. Since the number of participants who selected the first option was small, this option was combined with the second one. Thus, sleep quality was assessed on a 4-point scale. For sleep quantity, the participants were asked “How many hours do you usually sleep a day?” with the following four options: less than 4 h, 4–6 h, 6–8 h, and 8 h or more. The study also assessed the sleep score, reflecting both sleep quantity and sleep quality, against the development of coronary heart disease. Short sleep duration and poor sleep quality were found to be associated with an increased risk of coronary heart disease. Participants who had lower sleep scores also showed a higher risk of the disease. The authors concluded that both sleep quantity and sleep quality should be considered for developing strategies for improving sleep, with the aim of preventing coronary heart disease.

Moore et al. found that the effects on health of sleep quality, assessed by 5-point scale (1 = poor and 5 = excellent; mean value of 3.1 with a standard deviation of 1.2), were greater than the effects of sleep quantity, even after adjusting for other individual characteristics [25]. They also showed that the impact of sleep quality on physical health increased as the subjects slept for shorter durations.

Yang et al. [26] investigated the association between sleep measures (quantity and quality) and BMI among 10,007 adults living in the Philadelphia area. Sleep quality was assessed by asking about the participants’ sleep quality on a 5-point scale, with 1 being restless and 5 being restful. The question on sleep duration was “How many hours of sleep do you get at night?”. The overall sleep quality score was 3.61, with a standard deviation of 1.29. On average, men reported better sleep quality than women. Fifty-five per cent of the women and 53% of the men slept for at least 7 h every night; 27% slept for between 6 and 7 h, and 20% slept for less than 5 h. The study found that better sleep quality was related to lower BMI in women, while men who slept for less than 5 h had a higher BMI than those who slept for 7 h or more. In order to reduce BMI, it may be useful to help men to increase their sleep duration and women to improve their sleep quality. These authors also examined the effect of both sleep quantity and sleep quality in mediating the relationship between perceived discrimination and health, and found that sleep quality rather than sleep duration mediates the unfavorable effect [27].

A meta-analytic study demonstrated that both short and long sleep duration, and difficulty in maintaining sleep, were associated with a greater risk of type 2 diabetes [71]. From these results, Cappuccio et al. concluded that quantity and quality of sleep consistently and significantly predicted the risk of the development of type 2 diabetes. The risk varies between 28% in people who report habitual sleep of 5–6 h per night and 84% in those with difficulties in maintaining their sleep.

In patients with chronic kidney disease, Ricardo et al. reported that ESS, but not PSQI, was associated with an increased risk of all-cause mortality [72]. In addition, they found significant associations between greater sleep fragmentation and an increased risk of end-stage renal disease, and between both greater sleep fragmentation and shorter sleep duration and both a greater decline in estimated glomerular filtration rate and increased protein excretion. From these results, they concluded that short sleep, as well as poor sleep quality, are risk factors for the progression of chronic kidney disease.

Taking these reports into consideration, we should emphasize that more attention should be paid to obtaining good quality sleep (restfulness, no sleepiness, no need for more sleep, sufficient objective sleep depth, etc.) in terms of avoiding sleep-related health problems.

## 10. Sleep Quality in Children and Adolescents

So far, studies on sleep quality in children have been limited. As cited before, Falbe et al. [21] demonstrated that sleeping near a small screen was associated with a significantly poor sleep quality in children of grades 4 and 7. Bruni et al. [28] assessed the sleep quality of pupils aged 11–16 years by asking 15 questions on irregular sleep habits, prolonged sleep latency, and difficulties in waking up in the morning. Each question was answered on a 5-point scale. The authors found that evening circadian preference, mobile phone and Internet use, number of other activities (music, television, gaming console, sport, etc.) after 21:00, late turning off time, and number of devices in the bedroom had an unfavorable effect on sleep quality.

In addition to poor sleep quality related to these electric devices, Phillips et al. emphasized the presumable influence of sleep quality on health [73]. Indeed, Blackwell et al. [74] reported an association of good sleep quality with better general health.

Javaheri et al. [75] investigated the association between sleep variables (sleep quantity and sleep quality) and blood pressure in 238 adolescents aged 13–16 years who had no sleep apnea or severe co-morbidities. Actigraphic data was used, and a short sleep duration (6.5 h or less) and low sleep efficiency (85% or less) were used for the sleep variables. These authors found that the odds for prehypertension, which was defined as either systolic, diastolic, or both, blood pressure being at or above 90% of the age, gender, and height standardized value, was increased 4.5-fold among participants with low sleep efficiency, and 2.8-fold in those with short sleep duration. In addition, the participants with poor sleep quality (low sleep efficiency) were found to have a systolic blood pressure 4.0 mmHg higher than the other adolescents.

Martikainen et al. [76] examined the relationships between both sleep quantity and sleep quality and cardiovascular function in children with a mean age of 8.0 years who had no sleep-disordered breathing. Sleep data were obtained through Actiwatch, and sleep latency, sleep fragmentation, and sleep efficiency were used as variables reflecting sleep quality. These authors found that neither sleep quantity nor sleep quality showed significant associations with 24-h ambulatory blood pressure or cardiovascular reactivity.

Berentzen et al. [29] found that girls aged 11–12 years showed cardiovascular risk related to both sleep quantity and sleep quality, but they failed to obtain significant results for boys. They used a questionnaire to obtain sleep characteristics such as time in bed, bedtime, getting-up time, night-time awakenings, and daytime outcomes. The latter two issues were taken to reflect sleep quality. There were three questions on daytime outcomes. The first one was about “difficulty with getting up in the morning” and the second was about “feeling rested after waking on a school day”. On these questions, the answers were selected from two categories (yes or no). The third question was about the frequency of feeling sleepy or tired during the day, and the answers were divided into two categories, “never” or “sometimes” and “at least once a week”. Girls who spent a long time in bed (11–12.5 h) had a significantly lower body mass index and a smaller waist circumference compared with girls who spent 10–10.5 h in bed. Girls who went to bed late and rose early had significantly higher total cholesterol and higher high-density lipoprotein cholesterol than “early to bed/early to rise” girls. Girls who woke during the night almost every night, sometimes in combination with falling asleep again after a while or after a very long time, had higher total cholesterol than girls who did not wake during the night. Girls who felt sleepy/tired on one or more days per week had significantly lower high-density lipoprotein cholesterol and higher total cholesterol/high-density lipoprotein cholesterol ratio than girls who did not feel sleepy.

According to Komada et al. [77], a COVID-19-related school closure decreased sleepiness in children and was associated with decreasing social jetlag, although with a delay in bedtime. In general, late bedtime is considered to be an unfavorable habit [44]. The overall effects of these alterations have remained to be clarified.

Summarizing the associations between sleep quality and health or disorders in children and adolescents, a lot of issues were found to be remained to be investigated. In particular, no definite criteria for assessing sleep quality have been established. Of course, it may be difficult to take only one aspect of sleep quality from a standardized set of criteria, because sleep has many biological, psychological, and other unknown aspects. Further broad studies are necessary to discover the details of sleep quality, especially among children.

## 11. Another Sleep Issues to Be Investigated

This review lacked discussions on timing of sleep. Disturbance of sleep phase has been known to be associated with chronotype [78]. Chronotype is reported to be associated with sleepiness [79], sleep-onset duration and sleep efficiency during the first two years of life [80], and metabolic dysfunction [81,82]. Timing of sleep and chronotype are also important issues to be investigated in terms of both private and public health.

## 12. Conclusions

Both sleep quantity and sleep quality are fundamental components of sleep. Since there are marked inter-individual differences in the need for sleep, sleep quantity might be assessed not by actual duration but by a deviation from the individual’s own optimal sleep quantity. For sleep quality, no standardized scale has been established, probably because sleep quality has many aspects. The variables for sleep quality can be selected depending on the purpose of the study.

At present, many studies have decided that sleep quality, rather than sleep quantity, is favorable for reflecting health or functioning issues. However, these studies used actual values for sleep quantity. Further studies using the deviation from an individual’s own optimal sleep quantity may show us another aspect of the effects of sleep quantity on various life issues.

## Data Availability

Not applicable.

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
