# Peer review of "Which Is More Important for Health: Sleep Quantity or Sleep Quality?"

_children, 2021, doi:10.3390/children8070542_

Round 1
Reviewer 1 Report
Thank you for the opportunity to review: Which is more important for health: Sleep quantity or sleep quality?
The author provides an overview of sleep health posing the question of whether sleep quantity or sleep quality is a more important parameter associated with various health outcomes. While this was a thorough review, it may be beneficial to reorganize thoughts as to not confuse whether the author is referring to the youth population or adult populations. Consistency throughout would help with readability. Furthermore, there were many syntax errors noted throughout the manuscript making it difficult to follow author’s ideas. An editor to read it over and provide proper translation into English would greatly strengthen the manuscript’s readability.
Overall, there is room to discuss other factors such as sleep phase delay when speaking to adolescent sleep as it is well documented in the literature. Furthermore, although discussed briefly, a more thorough discussion on the chronotype would be beneficial.
The review is very broad that more subtitles may help to organize the review.
Some minor considerations:
In the First paragraph (lines 25-28) the author makes some assumptions about the perspectives of modern society on the importance of sleep. Please reference this statement, as more research is actually demonstrating that people value sleep, they may not prioritize sleep because of other competing interests (e.g., exercise, work, school, socialization).
In the second paragraph the author provides some historical elements of sleep duration in children specifically in Japan. Please provide how sleep duration was measured in this sample – is it parental reports? Or were objective measures used to determine sleep duration (e.g., actigraphy) – obviously there was no actigraphy in 1935, but in 2003 researchers started measuring sleep objectively.
Line 57-58 – consider re-wording – The way it read now, is that people are actively looking for ways to get less sleep duration and increasing their sleep quality. Can you please clarify this statement? Perhaps, it’s better to say, that in modern society, sleep is often not made a priority due to competing interests…. Furthermore, you may want to indicate that sleep duration is not the only sleep characteristic that defines healthy sleep, but sleep quality.
Line 59-60 – Instead of “As regards recent trends…” consider, “Regarding recent trends in US adult sleep duration, ….”
Lines 67-69 – consider changing to, “In spite of the aforementioned description of a sleep-deprived society [7], we are unable to confirm decreases in adult sleep duration since there is no data available from the early 1900s.
Lines 69-70 – this sentence feels contradictory – the author summarizes the literature on how sleep duration (i.e., sleep quantity) has declined and now is saying that it’s not sleep quantity that is said to be declining, but sleep quality?
Line 71 – you cannot begin a paragraph with “However”… please consider changing to “the importance of sleep quality has gained recognition as an important sleep characteristic much later than studies conducted on solely on sleep quantity.” Or something to that effect.
Line 100-101 – This is a strong assumption – perhaps, “The differences seen in sleep quantity may be contributed to cultural differences.”
Line 102-103 – Screen time and extra-curricular after-school activities have been identified as factors that contribute to the decrease in sleep quantity.”
Line 111 – indicate that the student population studied was in Japan. Can these results be generalized to other student populations? Were there any data in other countries that found similar results?
Line 145 – Consider changing to “The amount of sleep needed varies from person to person and from night to night.” There are a number of factors that influence sleep quantity including: genetic, behavioural , medical and environmental factors.”
Line 153 – 154 – the author attributes the minimal recommendation of sleep duration is what the population aims for. To substantiate this claim, please provide a reference – perhaps a qualitative study that looked at that?
Line 179 – Consider changing to “Thus, the SNI was designed to use the following formula….” Rather than “designated using…”
Line 227 – consider re-wording.
Author Response
For reviewer 1
The author provides an overview of sleep health posing the question of whether sleep quantity or sleep quality is a more important parameter associated with various health outcomes. While this was a thorough review, it may be beneficial to reorganize thoughts as to not confuse whether the author is referring to the youth population or adult populations. Consistency throughout would help with readability.
Thank you for your important comment. I explain the standpoint of this review in the line 111-113.
Furthermore, there were many syntax errors noted throughout the manuscript making it difficult to follow author’s ideas. An editor to read it over and provide proper translation into English would greatly strengthen the manuscript’s readability.
I tried my best to correct English with my native friends.
Overall, there is room to discuss other factors such as sleep phase delay when speaking to adolescent sleep as it is well documented in the literature. Furthermore, although discussed briefly, a more thorough discussion on the chronotype would be beneficial.
Thank you for comments. But timing of sleep and chronotype would be too wide to discuss in this review. I described these issues should be discussed in a future manuscripts. (lines 494-499)
The review is very broad that more subtitles may help to organize the review.
I made two more sessions entitled “Sleep quantity and quality in patients with autism spectrum disorder and attention deficit/hyperactivity disorder” and “Another sleep issues to be investigated” as the eighth and eleventh ones.
Some minor considerations:
In the First paragraph (lines 25-28) the author makes some assumptions about the perspectives of modern society on the importance of sleep. Please reference this statement, as more research is actually demonstrating that people value sleep, they may not prioritize sleep because of other competing interests (e.g., exercise, work, school, socialization).
According to author’s comment, I added a reference number 3, and focused the discussion on Japan.
In the second paragraph the author provides some historical elements of sleep duration in children specifically in Japan. Please provide how sleep duration was measured in this sample – is it parental reports? Or were objective measures used to determine sleep duration (e.g., actigraphy) – obviously there was no actigraphy in 1935, but in 2003 researchers started measuring sleep objectively.
According to author’s comment, a sentence was inserted on line 43-44.
Line 57-58 – consider re-wording – The way it read now, is that people are actively looking for ways to get less sleep duration and increasing their sleep quality. Can you please clarify this statement? Perhaps, it’s better to say, that in modern society, sleep is often not made a priority due to competing interests….
Thank you for giving me a suitable comment. I changed the description as you recommended.
Furthermore, you may want to indicate that sleep duration is not the only sleep characteristic that defines healthy sleep, but sleep quality.
Thanks. I tried to change the description to show what I want to say.
Line 59-60 – Instead of “As regards recent trends…” consider, “Regarding recent trends in US adult sleep duration, ….”
I changed the description as you recommended. Thank you.
Lines 67-69 – consider changing to, “In spite of the aforementioned description of a sleep-deprived society [7], we are unable to confirm decreases in adult sleep duration since there is no data available from the early 1900s.
I changed the description as you recommended, thanks.
Lines 69-70 – this sentence feels contradictory – the author summarizes the literature on how sleep duration (i.e., sleep quantity) has declined and now is saying that it’s not sleep quantity that is said to be declining, but sleep quality?
I would like to show you there is contradictory opinion.
Line 71 – you cannot begin a paragraph with “However”… please consider changing to “the importance of sleep quality has gained recognition as an important sleep characteristic much later than studies conducted on solely on sleep quantity.” Or something to that effect.
Thanks. I changed the description as you recommended.
Line 100-101 – This is a strong assumption – perhaps, “The differences seen in sleep quantity may be contributed to cultural differences.”
According to your comments, I changed the description.
Line 102-103 – Screen time and extra-curricular after-school activities have been identified as factors that contribute to the decrease in sleep quantity.”
Thanks again. I altered the description.
Line 111 – indicate that the student population studied was in Japan. Can these results be generalized to other student populations? Were there any data in other countries that found similar results?
According to your comments, I put several citations on the results.
Line 145 – Consider changing to “The amount of sleep needed varies from person to person and from night to night.” There are a number of factors that influence sleep quantity including: genetic, behavioural , medical and environmental factors.”
I changed the description as you recommended.
Line 153 – 154 – the author attributes the minimal recommendation of sleep duration is what the population aims for. To substantiate this claim, please provide a reference – perhaps a qualitative study that looked at that?
Thanks. I put the term “at least in Japan” and cited a reference number 3.
Line 179 – Consider changing to “Thus, the SNI was designed to use the following formula….” Rather than “designated using…”
I changed the description as you recommended, thanks.
Line 227 – consider re-wording.
I changed the description. Thanks.
Reviewer 2 Report
This article provides an overview of the different measures of sleep quality and quantity. It allows us to situate the demographic, social and behavioral factors influencing the two sleep indices. The medical, adaptive and cognitive repercussions also tend to open the debate on this public health issue. Comments are sorted by order of appearance in the text.
Line 45-58: Using a graph to facilitate reading of sleep times would be relevant.
Line 76: I think it is necessary to make the link between the diagnostic criteria in DSM-4 and DSM-5.
Line 92-93: The values seem to be reversed (18 and 16 and 6 and 7).
Line 117-118: I do not understand what the difficulties of modern society associated with increased sleep quality refer to. This seems to be poorly explained here.
Line 168: What clinic is this?
Line 321-328 : Further investigation of the relationship between psychiatric history and sleep disorders would have been interesting in section 8. The medical conditions are well developed.
Line 382 : Graphical summaries would be relevant to facilitate the reading of Part "Sleep quality in children and adolescents"
General : It would have been interesting regarding the quality of sleep of children/adolescents to address in more detail the relationships between neurodevelopmental disorders (especially ADHD and ASD where anomalies are well documented in the literature). Executive and attentional skills were correlated but in a more global way.
Author Response
For reviewer 2
This article provides an overview of the different measures of sleep quality and quantity. It allows us to situate the demographic, social and behavioral factors influencing the two sleep indices. The medical, adaptive and cognitive repercussions also tend to open the debate on this public health issue. Comments are sorted by order of appearance in the text.
Line 45-58: Using a graph to facilitate reading of sleep times would be relevant.
Thank you for your valuable comments. Taken together with the referee 3 ‘s comment, some numbers were deleted and shorten the descriptions.
Line 76: I think it is necessary to make the link between the diagnostic criteria in DSM-4 and DSM-5.
I described the link as your comment, thank you.
Line 92-93: The values seem to be reversed (18 and 16 and 6 and 7).
Thanks. I changed the descriptions.
Line 117-118: I do not understand what the difficulties of modern society associated with increased sleep quality refer to. This seems to be poorly explained here.
I deleted the description on the increase of sleep quantity, thank you very much.
Line 168: What clinic is this?
I added the description on the clinic.
Line 321-328 : Further investigation of the relationship between psychiatric history and sleep disorders would have been interesting in section 8. The medical conditions are well developed.
Thank you for your valuable comment. But this is the review paper for the sleep quantity and quality. I hesitate to put deep into the psychiatric history and sleep disorders.
Line 382 : Graphical summaries would be relevant to facilitate the reading of Part "Sleep quality in children and adolescents"
Although an important comment is made, I found it difficult to illustrate the contents of this section.
General : It would have been interesting regarding the quality of sleep of children/adolescents to address in more detail the relationships between neurodevelopmental disorders (especially ADHD and ASD where anomalies are well documented in the literature). Executive and attentional skills were correlated but in a more global way.
Thank you for important comments. Though short, I put a section on the sleep quantity and quality of patients with ASD and ADHD.
Reviewer 3 Report
Thank you for this very nice review.
I found this review well written, in particular parts 5 to 10. Maybe just be careful sometimes some sentences are a little bit long and we need to read two times to understand correctly the message.
- Introduction :
Line 34 : You talk about an increase of sleep duration in children in some countries (UK, Canada …), we would like to know how much time was gained approximatively ?
Line 45 to 53 : these few lines contain a lots of numbers; that’s not really friendly to read, maybe it would nice to keep a little bit less information and concentrate the message into important numbers only.
- Future assessment on sleep quantity
Some studies have worked using the difference of sleep time between weekdays and weekends as an indicator of sleep deprivation (with a threshold at 90 min). It might be interesting to have few lines about that.
Reutrakul, S., et al. Chronotype is independently associated with glycemic control in type 2 diabetes. Diabetes Care 36, 2523-2529 (2013).
Leger, D., et al. Short sleep in young adults: Insomnia or sleep debt? Prevalence and clinical description of short sleep in a representative sample of 1004 young adults from France. Sleep Med 12, 454-462 (2011).
Author Response
For Reviewer 3
I found this review well written, in particular parts 5 to 10. Maybe just be careful sometimes some sentences are a little bit long and we need to read two times to understand correctly the message.
Thanks. I tried to alter several long descriptions into short ones.
Introduction :
Line 34 : You talk about an increase of sleep duration in children in some countries (UK, Canada …), we would like to know how much time was gained approximatively ?
Thanks. I added the real numbers.
Line 45 to 53 : these few lines contain a lots of numbers; that’s not really friendly to read, maybe it would nice to keep a little bit less information and concentrate the message into important numbers only.
Thanks. Some numbers were deleted and shorten the descriptions.
Future assessment on sleep quantity
Some studies have worked using the difference of sleep time between weekdays and weekends as an indicator of sleep deprivation (with a threshold at 90 min). It might be interesting to have few lines about that.
Reutrakul, S., et al. Chronotype is independently associated with glycemic control in type 2 diabetes. Diabetes Care 36, 2523-2529 (2013).
Leger, D., et al. Short sleep in young adults: Insomnia or sleep debt? Prevalence and clinical description of short sleep in a representative sample of 1004 young adults from France. Sleep Med 12, 454-462 (2011).
Thank you very much for valuable comments. I cited Leger et al at this section, and also mentioned about social jetlag. Reutrakul’s paper was also cited as reference number 81.